# Integrative Medicine and Plastic Surgery: A Synergy—Not an Antonym

**DOI:** 10.3390/medicina57040326

**Published:** 2021-04-01

**Authors:** Ioannis-Fivos Megas, Dascha Sophie Tolzmann, Jacqueline Bastiaanse, Paul Christian Fuchs, Bong-Sung Kim, Matthias Kröz, Friedemann Schad, Harald Matthes, Gerrit Grieb

**Affiliations:** 1Department of Plastic Surgery and Hand Surgery, Gemeinschaftskrankenhaus Havelhoehe, Kladower Damm 221, 14089 Berlin, Germany; fivos.megas@gmail.com (I.-F.M.); d.tolzmann@live.de (D.S.T.); jacqueline.bastiaanse@havelhoehe.de (J.B.); 2Department of Plastic Surgery and Hand Surgery, Burn Center, University of Witten/Herdecke, Kliniken der Stadt Köln, Ostmerheimer Str. 200, 51109 Köln, Germany; FuchsP@kliniken-koeln.de; 3Department of Plastic Surgery and Hand Surgery, University Hospital Zurich, Rämistrasse 100, 8091 Zurich, Switzerland; bong-sung.kim@usz.ch; 4Institute of Integrative Medicine, University of Witten/Herdecke, Alfred-Herrhausen-Straße 50, 58448 Witten, Germany; matthias.kroez@klinik-arlesheim.ch; 5Research Department Klinik Arlesheim, Pfeffingerweg 1, 4144 Arlesheim, Switzerland; 6Research Institute Havelhoehe, Kladower Damm 221, 14089 Berlin, Germany; friedemann.schad@havelhoehe.de (F.S.); harald.matthes@havelhoehe.de (H.M.); 7Institute of Social Medicine, Epidemiology and Health Economics CCM, Charité University Medicine, Charitéplatz 1, 10117 Berlin, Germany; 8Department of Plastic Surgery and Hand Surgery, Burn Center, Medical Faculty, RWTH Aachen University, Pauwelsstrasse 30, 52074 Aachen, Germany

**Keywords:** integrative medicine, complementary medicine, plastic surgery, anthroposophic medicine

## Abstract

Background: Integrative medicine focuses on the human being as a whole—on the body, mind, and spirit—to achieve optimal health and healing. As a synthesis of conventional and complementary treatment options, integrative medicine combines the pathological with the salutogenetic approach of therapy. The aim is to create a holistic system of medicine for the individual. So far, little is known about its role in plastic surgery. Hypothesis: We hypothesize that integrative medicine based on a conventional therapy with additional anthroposophic therapies is very potent and beneficial for plastic surgery patients. Evaluation and consequence of the hypothesis: Additional anthroposophic pharmacological and non-pharmacological treatments are promising for all areas of plastic surgery. We are convinced that our specific approach will induce further clinical trials to underline its therapeutic potential.

## 1. Introduction

The concept of integrative medicine (IM) combines conventional and complementary medicine and has become an increasingly emerging area of interest for patients and professionals alike. The regulations of complementary medicine are constantly developing, and the professional licensing and health insurance programmes are multiplying rapidly [1,2,3,4,5]. While the conventional understanding of medical treatment was predominantly in reacting to pathologic values, modern integrative medicine rather aims at promoting salutogenetic and hygiogenetic health as sources in a proactive manner [6].

IM in the context of anthroposophic medicine focuses on the human being as a whole—on the body, vitality, mind, and spirit—to achieve optimal health and healing [6]. The most widely used approach to complementary medicine can be divided into two subcategories: natural products, as well as mind and body practices and acupuncture [7]. This includes appropriate therapeutic and lifestyle approaches as well as healthcare professionals and most modern disciplines.

As a synthesis of conventional and complementary treatment options, IM combines the pathogenetic with the salutogenetic or hygiogenetic approach of therapy [6]. The aim is to create a holistic system of medicine for the individual. Precisely this concept addresses the patient’s needs and requirements of the present; a period of time in which self-determination and personal responsibility have become more and more important [8].

For example, in anthroposophic integrative oncology, it is an established concept to propose mistletoe therapy (*Viscum album* L., VA) concomitant to antineoplastic treatment in cancer patients in order to improve the tolerability of oncology-induced toxicity [9]. In anthroposophic cardiology, for example, the influence of rhythmic massage on heart rate variability has been studied [10].

The field of complementary medicine, as far as natural products are involved, is already present in plastic surgery [11]. For example, *Arnica montana*, onion extract, Vitamin E products, and *Melitolus* are considered beneficial [11]. Mind and body practices, such as hypnosis and meditation, are also known to have a positive effect on the postoperative course of plastic surgery patients [11]. It should also be mentioned that therapeutic concepts of integrative medicine can be applied in all four pillars of plastic surgery (aesthetic surgery, reconstructive surgery, burns, and hand surgery) [2,11,12,13,14]. However, little is known about the impact of a broader integrative concept that is based on anthroposophic medicine and combines these individual offerings and procedures, as the utilisation of IM in plastic surgery departments seems to be under-frequented [2]. We aim to share our encouraging first experiences of anthroposophic medicine integrated into standard plastic surgery treatments.

## 2. Hypothesis

We hypothesize, that integrative medicine based on a conventional therapy with additional anthroposophic therapies is potent and beneficial for healing and post-operative recovery of plastic surgery patients.

As a general hospital, the “Gemeinschaftskrankenhaus Havelhoehe (GKH)” provides a broad spectrum of surgery with an integrative approach [15]. The demand of complementary therapy concepts next to conventional medicine leads to a very attractive and healthy environment at the GKH, even for patients with an international background.

The newly established department of plastic surgery (2016) offers a broad spectrum of treatments, which ranges from reconstructive surgery, breast surgery, hand surgery, and burn surgery to aesthetic surgery. Our department pays special attention to continuing the extraordinary offer of integrative medicine, which was introduced in our breast cancer center (BCC) by the gynaecological and oncological departments [15].

During in-patient stay, complementary treatment options are offered, such as anthroposophical massage (rhythmical massage and streaming massage), breathing therapy, ergotherapy, eurythmy therapy, hyperthermia, painting therapy, clay modelling therapy, music therapy, physiotherapy, and psychotherapy. Therefore, not only functional and physical approaches are stressed, but also the mental state, creativity, and self-determination are focused on and promoted. Next to these mind body practices, anthroposophical care therapy and natural products are provided, such as *Arnica montana* salve for haematoma, *Bryophyllum pinnatum* for calming, aroma oils for wellbeing, and *aurum*/*lavendulan*/*rosae* unguents against agitation with tachycardia and (cardiac) restlessness.

The described IM concept is regularly integrated into the daily clinical routine and individually adapted to the wishes of the patients, as every patient has their individual requirements and expects individual support: for example, patients with chronic wounds walk through an exhaustingly long hospital stay and additionally expect variety and social assistance. Hand-surgery patients with functional impairments also need physical and interactive challenges and oncological patients with a life-shortening prognosis certainly require mental encouragement.

In this article, we hypothesize that integrative medicine based on a conventional therapy with additional anthroposophic therapies is potent and beneficial for plastic surgery patients, leading to faster wound healing, less complications, and a more effective treatment for the body as a whole, also including the mind and spiritual dimensions.

Over a period of three years (2018–2020), about 150 patients have received the anthroposophic program and complementary therapies in our plastic surgery department at Havelhoehe. These patients have accepted and fully completed the offered program for at least six days. Specifically, the program included as standard: eurythmy therapy, music therapy, painting therapy, rhythmic massage, physiotherapy, psychoeducation, a biographical interview, and the application of aromatic wet packs and embrocations (with, e.g., *thyme* or *aurum*/*lavendulan*/*rosae*). The heterogeneous patient population included cases with chronic wounds, extensive skin soft tissue defects, hand surgical clinical pictures, acute burns, as well as burn complications, but also aesthetic operations. Based on the feedback from these patients, we have developed our hypothesis.

## 3. Evaluation of the Hypotheses

Broadly speaking, despite the experienced demand for complementary therapy methods in plastic surgery [2,13]. The first publications mentioning plastic surgery in combination with complementary medicine was published in the 1970s [16]. However, as already reported, the current published data on these aspects are sparse. Still, the positive effects on the emotional state of the patients and the perception and processing of a perioperative and postoperative course are indisputable [17,18,19]. Surveys have even shown that patients expect plastic surgeons to be familiar with integrative medicine [13,20]. One of these surveys, by Patel et al., described that 80% of plastic surgery patients received integrative medicine services, such as natural products and mind–body practices. The majority of the questioned patients (71%) strongly believe in self-healing [13]. This elective surgery population seeks and examines the possibility of complementary treatments [21]. Patients undergoing aesthetic surgery are reported to have a high incidence of psychosocial issues [22,23]. A department offering a surgical therapy in combination with a holistic approach that stresses the importance of mental wellness is promising. However, so far, no studies have been performed with respect to treatment based on anthroposophic medicine.

If applied, the effects of natural products and of course potential complications have to be monitored in a standardized way. Possible complications of natural products are, for example, postoperative bleeding, hypertension, and dry eyes [24,25,26]. The use of herbals like *Gingko* caused bleeding in a blepharoplasty patient and a perioperative therapy with acupuncture was associated with a risk of wound-site infections in a lipoplasty patient [27,28]. Taking into account this important information, we believe that all integrative approaches, which have been described so far mainly in aesthetic-surgical patients, can be applied to all areas of plastic surgery [11,27].

In the US, nearly two thirds of medical schools have incorporated courses and/or clerkships in complementary and alternative medicine and the National Center for Complementary and Integrative Health states that 40% of the American population utilizes IM concepts [1,29]. In Germany, a cross-sectional survey of a nationally representative sample of women and men aged 18–69 years was conducted, which showed that overall the effectiveness and usefulness of natural therapies was positively evaluated by the majority of the study population and 58% of those surveyed would like to see such therapies prescribed more often [30].

In daily clinical practice, IM is used to relieve symptoms associated with chronic or terminal illnesses or side effects of conventional treatments [31]. In pain therapy, complementary, non-pharmacologic approaches show a positive effect and have already begun to replace some of the conventional medication in times of opioid epidemics [31].

Therapies such as acupuncture, massage, biofeedback, and natural remedies help to manage pain and limit reliance on opioids. Although the literature so far indicates that modern integrative in-patient treatment is mostly cost-equivalent to conventional treatment therapies, there could also be a favourable economic aspect here, saving costs for opioids by weaning patients off them early [32,33,34,35].

In oncology, the concept is not only based on curing, prolongation of life, or symptom relief, but also on the enhancement of physical, emotional, and or spiritual well-being, as well as maintenance of control over cancer and its treatment [15]. In this context, the addition of mistletoe (*Viscum album*) to targeted therapy significantly reduced the probability of adverse-event-induced oncological treatment discontinuation by 70% [9]. In addition, transcendental meditation can lead to a higher quality of life in older breast cancer patients [36].

As the meta-analysis by Hole et al., shows, music, which we regularly offer in our clinic, is extremely helpful perioperatively. In the 73 randomised controlled trials included in the study, the choice of music, timing, and duration varied. However, it was found that postoperative pain and analgesic consumption were lower and patient satisfaction was higher than in the control group without perioperative music [37].

It is further reported that there is a beneficial use of *Arnica montana* to decrease postoperative oedema and ecchymosis after rhinoplasty, as well as a positive impact of onion extract on improving scar pigmentation, hypnosis approaches to alleviate perioperative anxiety, as well as acupuncture as a methodology to improve perioperative nausea [14,34,38,39,40,41,42].

While IM centres are becoming more established, the referrals to these centres in the US (02/2017) were analysed by Ruan et al. In total, 73.8% of the patients were primarily referred from departments of medicine; only 13.1% were from the departments of surgery. From this 13.1%, merely 0.77% (0.077% of all referrals) were primarily referred from the department of plastic and reconstructive surgery [2]. These facts also suggest the assumption that even in hospitals where IM is offered, it is rarely applied by plastic surgery departments. Because of this, we would like to emphasise our hypothesis that plastic surgery in combination with IM based on anthroposophic medicine is a promising treatment.

A large part of the daily work of our department concerns chronic wounds. The risk factors for such wounds are manifold. Systemic risk factors, such as malnutrition, obesity, vascular disease, diabetes mellitus, cancer, immunosuppression, and personal habits (e.g., smoking), favour the detrimental conditions as much as local risk factors such as neuropathy, local pressure, repetitive trauma, or radiation [43]. Optimised health behaviours would have a positive effect on many of these points, as some studies have already shown [44,45]. Therefore, in our hospital we try to support patients in this respect, to give them an impulse or to reinforce them with our holistic offer.

Even though the demand for IM and the utilisation in plastic surgery is obviously high, to the best of our knowledge, there are no other hospitals in Germany that provide IM in their departments of plastic surgery and offer complementary therapy options aiming at a holistic approach.

The Gemeinschaftskrankenhaus Havelhoehe found a way to include anthroposophic medicine in plastic surgery during in-patient time. This synergistic concept meets the requirements of the present and empowers the patient’s sanitary self-determination and holistic concept.

Nevertheless, the limitations of a complementary approach should be mentioned. In many cases, for example, the dosages of the prescribed medications are an empirical value of the physician and are not suggested in a guideline. Further, there is a huge lack in terms of randomized trials, especially in the field of general or plastic surgery. To date, there is no randomized study dealing with plastic surgery and complementary medicine or anthroposophic medicine. However, some studies concerning chronic pain or diabetic polyneuropathy have surgical aspects [46,47,48]. Expanding the included literature reveals a variety of negative and positive studies related to integrative therapy approaches and the corresponding issues investigated [46,49,50,51,52,53].

## 4. Consequence of the Hypothesis

Our patients report that they have perceived our integrative therapies very positively. In addition, plastic surgery departments can increase their attractiveness and patient satisfaction in general by offering IM. Finally, an economic factor not to be neglected could be present, for example, through earlier weaning off pain medication. Thus, we are convinced that our approach will provide a strong impulse for further randomized controlled trials, which are definitely necessary in this area.

## Data Availability

Not applicable.

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
