# Peer review of "Integrative Medicine and Plastic Surgery: A Synergy—Not an Antonym"

_medicina, 2021, doi:10.3390/medicina57040326_

Round 1
Reviewer 1 Report
Thorough review of the literature.
Author Response
We are glad that Reviewer 1 finds the manuscript principally suitable for publication in the journal, and thank the reviewer for outstanding constructive criticism.
As suggested, we performed the review of literature even more thorough.
Reviewer 2 Report
The authors have highlighted a very relevant topic in regards to the current scenario, and I believe the content will be of profound interest to the readers.
I, however, suggest some revisions, which I believe will improve the scientific rigor.
The authors should add a section on the cons of using integrative medicine. For example, studies with regards to IM are limited, and the scientific research is lacking on the utility of using IM, no information regarding dosing in many cases, government approvals and so on.
Alongside, I suggest some in-depth literature review on IM used in plastic surgery or some specific examples of IM and how they could be used. The writeup could be backed up by in vitro or in vivo or clinical studies performed (if any).
I suggest the authors add another section explaining how the conventional medicine can complement the integrative medicine, by adding some examples specifically relating to plastic surgery.
Author Response
We are glad that Reviewer 2 finds the manuscript principally suitable for publication in the journal, and thank the reviewer for outstanding constructive criticism.
As suggested, we added a section on the cons and limits of using integrative medicine.
We further expanded our cited literature and specifically focused on randomized controlled trials (which up to now do not exist – at least concerning plastic surgery in combination of complementary medicine).
In addition, we added a description of our program that is offered to the patients at our hospital, revealing different examples, as suggested.
Round 2
Reviewer 2 Report
Most of my comments have been resolved, and the manuscript is now suitable for publication.